# Antimalarial Drug Strategies to Target *Plasmodium* Gametocytes

**Bruce A. Munro and Brendan J. McMorran ***

Division of Immunology and Infectious Disease, The John Curtin School of Medical Research,
College of Health and Medicine, The Australian National University, Canberra, ACT 2601, Australia;
bruce.munro@anu.edu.au
* Correspondence: brendan.mcmorran@anu.edu.au

**Abstract:** Onward transmission of *Plasmodium falciparum* from humans to mosquitoes is dependent upon a specialised transmission stage called the gametocyte. Despite its critical role in transmission, key questions regarding gametocyte biology remain to be answered, and there are no widely prescribed therapeutics to eliminate them. Advances in our understanding of the biology of the gametocyte in combination with growing information regarding the mechanism of action of anti-plasmodial therapies provide an emerging view as to which of the biological processes of the gametocyte present viable targets for drug intervention and explain the variable activity of existing therapies. A deeper understanding of the gametocyte and transmission stages of *P. falciparum* is a path to identifying and characterising novel drug targets. This review will examine how a selection of current and potential gametocytocidals mediate their effect.

**Keywords:** *Plasmodium*; gametocyte; antimalarial drug; gametocidal drug targets

## 1. Introduction

Malaria remains a leading global infectious disease and, especially in the developing world, a major cause of morbidity and mortality. According to estimates from the World Health Organisation (WHO), globally in 2020, there were approximately 240 million cases of malaria causing over 600,000 fatalities, most of whom were young children under age five [1]. The cause of malarial disease is infection of the host by the apicomplexan parasite *Plasmodium.* Of five species infective of humans, *Plasmodium falciparum* is responsible for the vast majority of the world's malaria burden [1,2].

Human infective *Plasmodium* requires a primary host, an *Anopheles* mosquito, and a secondary host, a vertebrate, to complete its lifecycle. To facilitate the infection of and transmission between these two hosts, *Plasmodium* has developed uniquely adapted parasitic forms that are responsible for transmission between vertebrate and mosquito. These transmission stages create key bottlenecks within the parasite lifecycle [3,4]. Current and historic malaria control initiatives have relied upon the widespread use of insecticide-treated nets as vector control, and concurrent diagnostic screening and antimalarial drug administration [5,6]. The majority of successfully prescribed antimalarial therapies are directed against the proliferative and symptom-causing asexual blood stage of *Plasmodium* [7]. These measures have succeeded in bringing cases and fatalities to new lows but have been marked by a legacy of insecticide and antimalarial resistance [8,9]. Global estimates suggest that progress against the disease is stalling, and if the ambitious goal of malaria eradication, either globally or locally, is to be achieved, therapies with potent activity against the transmission stages of the parasite are considered an unmet but important need [10,11]. As the only parasitic stage capable of transmission from vertebrate to mosquito, the gametocyte represents a rational choke point for the deployment of transmission blocking therapies. Although not contributing to disease symptoms, it is the onward transmission of gametocytes that underpins the spread of antimalarial resistance and enables genetic recombination during sexual reproduction [12]. Understanding the complex and diverse

biological processes underpinning gametocyte growth and transmission will help to explain the relative effectiveness of current antimalarials and reveal new strategies for developing gametocytocidal therapies.

### 1.1. The P. falciparum Lifecycle

In humans, a *P. falciparum* infection begins when sporozoites are injected into the skin during a blood meal by an infected female *Anopheles* mosquito. Within minutes, these parasites home to the liver and invade hepatocytes, where they proliferate to produce thousands of merozoites [13]. Following their release into the bloodstream, the merozoites invade erythrocytes, initiating the asexual blood stage (ABS) of the lifecycle.

The asexual stage infection is characterised by parasite growth to enable a cyclic pattern of successive invasion of, and egress from, circulating host erythrocytes. A *P. falciparum* ABS cycle lasts approximately forty-eight hours, progressing through morphologically distinguishable ring, trophozoite, and schizont stages, which replicate to produce 8–36 daughter merozoites. Erythrocyte rupture releases the merozoites to the circulation and is responsible for most clinical symptoms of malaria [14]. Cytoadherence of the parasitised erythrocytes to the vasculature additionally contributes to the pathology of complicated malaria, especially that of cerebral malaria [15].

During the asexual stage of infection, ~1% of parasites commit towards development of the gametocyte, the sexually dimorphic, transmission capable stage of *Plasmodium* [16]. Factors driving the commitment towards gametocytogenesis are varied but potentially include signals from the host, other parasites, and population level pressures, although our current understanding of these processes is incomplete [17–19]. Commitment is initiated in the preceding generation of parasites, producing daughter merozoites that all commit to gametocytogenesis [20]. Commitment can also occur in the ring stage, leading to direct gametocyte differentiation without the need for an intervening asexual cycle [21]. Gametocytes are haploid and sexually dimorphic, with a pronounced female-to-male sex bias [22]. Sex determination is not determined by sex chromosomes but instead by differential gene expression during development, and occurs at the same time or shortly after commitment; as such, gametocyte-committing schizonts will produce either all male or all female cells [23]. Based on observations of other genera of the order Haemospororida, where asexual replication occurs within solid tissues, and only transmission capable forms infect erythrocytes, it has been suggested that gametocytogenesis commitment in *Plasmodium* occurs by default, and that active signalling is required to maintain erythrocytic asexual replication [24,25].

*P. falciparum* gametocyte development to a transmission-capable form occurs over a period of ten to twelve days and is classically divided into five morphologically distinct developmental stages, termed stages I–V [26]. Following invasion by committed merozoites, parasitised cells home to the extravascular spaces of the bone marrow where gametocyte develops occurs [27,28]. The mature stage V gametocyte re-enters the circulation, where it is available for ingestion by a feeding mosquito and has a lifespan ranging from several days to up to three weeks [29]. The sexual dimorphism of mature gametocytes is represented in the differential expression of hundreds of genes and proteins by males and females and is critical to their distinct roles in transmission and sexual reproduction [30].

Once ingested by the mosquito, the gametocyte immediately responds to the change in the environment of the insect midgut to initiate a series of rapid and controlled differentiation steps termed gametogenesis that produce haploid male (micro-) and female (macro-) gametes. Stimuli of gametogenesis include a drop in temperature of >5 degrees Celsius, a rise in pH, and the insect metabolite xanthurenic acid [31,32]. Male gametogenesis (exflagellation) produces eight flagellated microgametes through a process requiring three rounds of genomic replication and cell division, and extensive cellular remodelling and axoneme assembly [33]. Female gametogenesis is more restrained, producing a single macrogamete that enlarges to adopt a spherical morphology following its escape from the erythrocyte [34]. The motile microgamete actively seeks out the macrogamete within the midgut, and fertilisation occurs after binding and membrane fusion [35]. Fertilisation produces a single diploid zygote, which undergoes one round of DNA replication without cell division to become tetraploid [36]. Over the next 24 h, the initially round zygote will develop into a motile, elongated stage termed the ookinete, which is capable of a motility that enables it to penetrate the gut wall and escape to the haemocoel space [37]. Here, the ookinete encysts and undergoes a development and growth phase, termed sporogony, in which multiple rounds of parasite replication produce thousands of haploid nuclei that develop into sporozoites [38]. At the end of this replicative process, the oocyst ruptures, and the sporozoites are released, prepared to migrate to the salivary glands of the insect in anticipation of infection of a new human host [39].

There are strong selective pressures acting on the parasite during the transmission from human to mosquito. During a natural infection, of the thousands of gametocytes that a mosquito might ingest, only 50–100 will develop into ookinetes, and only 5 might successfully develop into oocysts [40,41]. These steps, therefore, create a bottleneck in the parasite lifecycle, which is critical to its onward transmission and may be uniquely vulnerable to antimalarial strategies.

*1.2. Gametocyte Biology*

*P. falciparum* biology, including gametocyte development and maturation, have been extensively characterised by genomic [42,43], transcriptomic [44,45], proteomic [46–48] and metabolomic studies [49]. These have revealed the dynamic changes that occur during the development of immature stage I–IV gametocytes to the mature stage V form and provide further evidence that distinguishes gametocytes as highly specialised compared to their ABS counterparts [50].

For example, the metabolic needs of ABS parasites needed to sustain multiple rounds of replication are met by high rates of host haemoglobin catabolism, nutrient uptake from the circulation, and nucleotide synthesis and salvaging [51]. Immature gametocytes have similar metabolic needs to the ABS in terms of utilising host haemoglobin, synthesising RNA, and transcribing new proteins; however, upon maturation of the stage V gametocyte, haemoglobin catabolism ceases, and upon re-entering the circulation, they enter a state of metabolic quiescence, with energy needs perhaps reduced to maintaining homeostasis [49,52,53]. Although this divergence between the ABS and gametocyte stages underlies the relative inactivity of most current antimalarials against gametocytes, for example, those that target haemoglobin digestion or nucleotide biosynthesis, such differences may also present the opportunity for novel mechanisms to block transmission [54,55].

### 1.3. Gametocytocidal Compounds

Few compounds with reliable gametocytocidal activity have been described and currently only primaquine is recommended by the WHO for clinical use, based on its ability to kill all stages of gametocytes [56]. Other existing antimalarials developed for treating the ABS show only incidental activity against the gametocyte, and as a consequence, gametocytes are refractory to most drug classes [57,58]. Some studies have reported that exposure to antimalarial treatments results in increased commitment to gametocytogenesis [59]; however, this remains to be definitively proven [59,60]. With the advent of high-throughput screening (HTS) protocols for assaying gametocyte responses to compound exposure, a swath of new compounds with gametocytocidal and transmission blocking activity have been identified, both from screens of the Medicines for Malaria Venture (MMV) library of compounds [61–63] and of other diverse library sets [64–66].

By combining advances in understanding gametocyte biology with the activity of compounds from HTS and their known or predicted mechanism of action (MOA), an understanding of how and where a compound mediates its effect can be established. In assaying for gametocytocidal activity, the metabolic quiescence of the mature gametocyte [67,68] presents challenges in determining their viability and in making comparisons between methodologies [69,70]. A range of protocols to study the transmission effects of drug-exposed gametocytes have now been developed, including for assaying sex-specific gametogenesis effects [71] and ookinete development via numerous in vivo and in vitro *Plasmodium berghei* mouse models of malaria [53,72,73].

The current "gold standard" in assaying transmission biology remains the standard membrane feeding assay (SMFA), a technically demanding protocol that sees drug-exposed gametocytes fed to *Anopheles* mosquitoes [74]. The dissection of the mosquito midgut and a count of the resulting oocysts provides a measure of successful transmission of the parasite, which notably produces a greater oocyst intensity than what is observed following a natural infection [75,76]. An alternative end point is to dissect the salivary glands of the mosquito to reveal a count of sporozoites, which reveals all of the transmission events from gametocyte uptake to sporozoite migration. Within the SMFA, a direct drug exposure (where a gametocyte culture is exposed to a compound shortly prior to mosquito feeding) and an indirect drug exposure effect (where a compound is incubated alongside a gametocyte culture for an extended period prior to feeding) and drug washouts prior to feeding can be used to interrogate which stages of sexual development are affected [72,74]. These assays have been further developed to incorporate the use of bioluminescent parasite lines to aid detection of transmission success [74].

Understanding the MOA of novel and existing antimalarial compounds is critical to determining resistance mechanisms and for informing future drug programs [77,78] and will be used in this review to relate the MOA to the reported activity against the gametocyte. Understanding how drug activity relates to the tightly controlled mechanisms of gametocytogenesis and mosquito stage development will produce an understanding of why some treatments fail and focus interest on areas of gametocyte biology which are prone to manipulation by drug administration.

In this review, we highlight examples of current antimalarial therapies that have variable utility at preventing gametocyte growth or onward transmission, and a selection of those which are known for their gametocytocidal activity. We then examine potential gametocyte-specific and transmission stage processes that may serve as novel therapeutic targets (Figure 1).

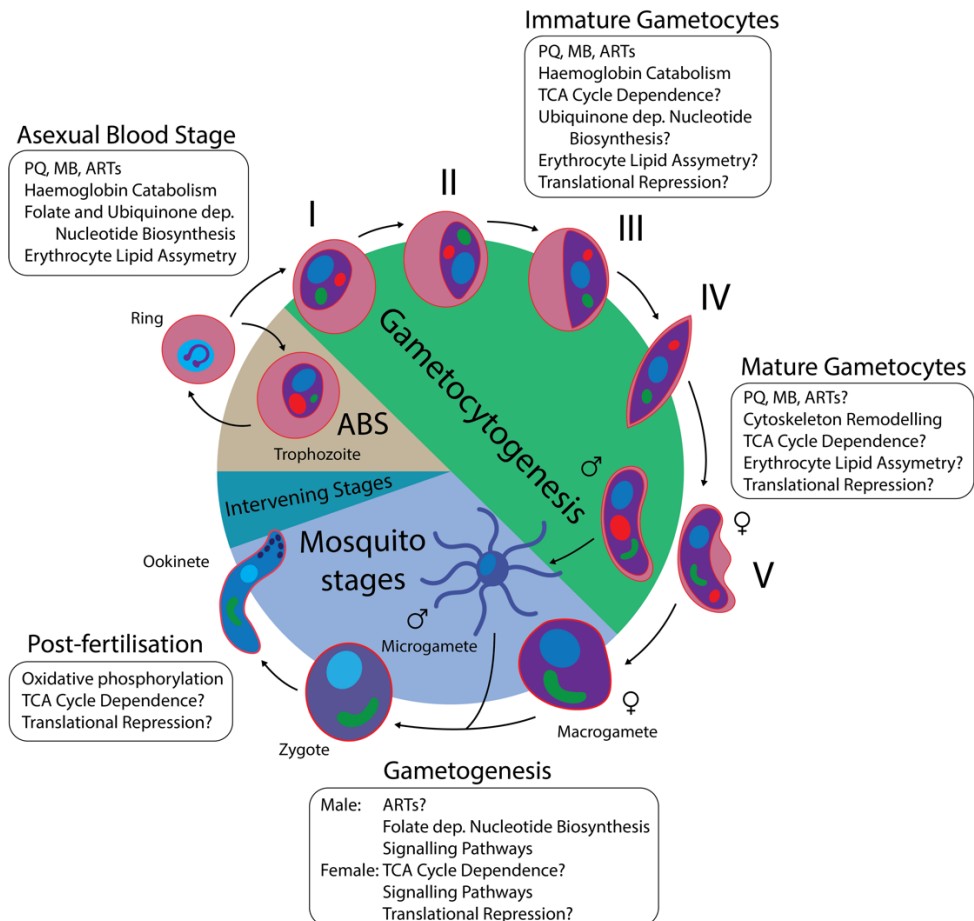

**Figure 1.** Targeting the gametocyte and transmission stages of *P. falciparum*. This overview highlights various stages of the parasitic lifecycle and their accompanying molecular processes that could be inhibited by compounds that are directly or indirectly active against the gametocyte. Included are the asexual blood stage (ABS), immature and mature gametocyte stages (I–V), and the mosquito stages including male and female gametogenesis, and the post-fertilisation stages; the zygote and ookinete. Blue: nucleus, Red: DV, Green: mitochondria. PQ (primaquine), MB (methylene blue), ARTs (artemisinin derivatives), "?" signifies potential of targeting this process.

## 2. Antimalarials with Variable Gametocytocidal Activity

*Haemoglobin Catabolism*

Host haemoglobin (Hb) provides a source of amino acids for the intraerythrocytic stages of *P. falciparum.* To access host Hb, parasites endocytose vesicles of erythrocyte cytoplasm and direct these towards an acidic and proteolytic organelle termed the digestive vacuole (DV). Here, Hb is digested, and small peptides are exported to the parasite cytoplasm, while the otherwise toxic heme liberated during this process is converted to an inert, nontoxic crystalised pigment termed hemozoin [79,80].

Some of the first antimalarials developed in the modern era act on haemoglobin catabolism. Chloroquine, a 4-aminoquinoquinoline, inhibits the conversion of heme to hemozoin by complexing directly to heme [81,82], and the consequent accumulation of free heme results in the death of the parasite. The actions of chloroquine are therefore closely linked to the rate of Hb catabolism. During the asexual growth phase, approximately 80% of host Hb is digested within the 48 h it takes to develop and mature to schizonts [83]. In contrast, the slower developing gametocyte consumes 50% of host Hb by stage II/III and 70% by stage IV, but further digestion is absent from stage IV parasites [84]. These findings provide an explanation for the activity of chloroquine against ABS parasites and immature gametocytes but not against mature gametocytes [64,85,86]. Other 4-aminoquinoquinolines

including piperaquine and amodiaquine, which appear to act differently to chloroquine in their inhibition of hemozoin formation, have been investigated as gametocytocidals. Piperaquine kills stage II/III gametocytes but not stage V in vitro [64,87], or when deployed in clinical trials as a monotherapy [88]. It does, however, contribute to increased gametocyte clearance when paired with ARTs [89,90]. Amodiaquine shows similarly activity only against immature gametocyte stages in vitro [87], and in clinical data, its effects appear to be limited [91,92]. Interestingly, amodiaquine is reported to inhibit male exflagellation in vitro, which may represent an independent MOA or the resumption of low levels of Hb digestion [53].

## 3. Antimalarials with Known Gametocytocidal Activity

### 3.1. Primaquine

Primaquine, an 8-aminoquinoline derivative, is currently the only antimalarial recommended by the WHO as a gametocytocidal therapy. First approved for use in 1952, it has been conclusively shown experimentally and clinically to directly kill gametocytes, interrupt transmission, and clear circulating gametocytes [93]. Primaquine has not, however, seen widespread use because of contraindications in people deficient in the enzyme G6PD, a genetic condition prevalent in many populations residing where malaria is endemic [94]. The MOA of primaquine against ABS and gametocyte stage parasites requires metabolism by the liver into numerous metabolites, the identities and activity of which are still under investigation [95].

A two-step mechanism has been proposed in which primaquine is first converted to hydroxylated metabolites by the host, which then undergo spontaneous oxidation to quinoemines with accompanying production of $H_2O_2$ [96]. Thus, primaquine activity against gametocytes and the ABS occurs via a relatively nonspecific MOA. The related compound tafenoquine might represent a future alternative to primaquine due to an improved safety profile [97,98].

### 3.2. Methylene Blue

Methylene blue is a synthetic dye first administered for use against malaria in the 19th century in Germany and saw general widespread use until the introduction of more effective therapies such as chloroquine [99]. It also has several undesirable side effects, producing blue urine and iris discolouration, and is contraindicated in G6PD-deficient individuals [100], although there has been recent interest in redeveloping the compound [101]. Methylene blue's potent activity against all stages of gametocyte development and against male and female gametogenesis in vitro is well described [64,71,73,87], and the compound potently blocks transmission in the SMFA [74,87]. Methylene blue reliably clears circulating gametocytes in clinical trials when paired to partner drugs or as a monotherapy [93,100,102].

The MOA of methylene blue against malarial parasites is poorly understood, though it may be multifactorial [100]. One hypothesis relates to the compound's mechanism in treating methaemoglobinaemia [103], a disease caused by elevated methaemoglobin levels in the blood. This mechanism sees methylene blue activity dependent on its ability for redox cycling, where upon exposure of methylene blue to NADPH, it is converted to leucomethylene blue in parasitised cells. Leucomethylene blue subsequently undergoes rapid autooxidation, producing $H_2O_2$ in the process [104], thus killing both ABS and gametocyte stage parasites in a nonspecific manner similar to primaquine. Other hypotheses of the MOA similarly relate to the compound's ability to distort the redox balance of the cell, including through synergistic effects with other antimalarials [105,106].

### 3.3. Artemisinin Derivatives

Artemisinin derivatives (ARTs) are the current global standard of treatment for uncomplicated malaria [56]. ARTs are typically administered as prodrugs that are converted to the active form dihydroartemisinin (DHA) by the cleavage of their endoperoxide bridge by free heme in the parasitised cell [107]. Once cleaved, a radical complex is formed, which causes nonspecific protein and lipid damage, eventually leading to parasite death [108]. Notably, free heme is released during Hb digestion, and as such, ARTs are most effective against the ABS and immature gametocytes, while contradictorily, several studies show activity against parasite stages where Hb digestion is accepted to have ceased, namely the stage V gametocyte [73,85,87,109], male exflagellation [71,87], and subsequent oocyst formation [87,110]. Some studies, however, have found no activity of ARTs against the stage V gametocyte [109,111].

These in vitro findings do not appear to translate into measurable clinical effects; the ART-induced reductions in gametocyte carriage rates (measured as up to four fold in some studies [29]) are instead attributable to elimination of the ABS [112]. These discrepancies may be because in vitro studies cannot adequately capture drug pharmacokinetic effects, particularly since the half-lives of ARTs are relatively short (hours) compared to the time spanning gametocyte development (days) [109]. However, while ART treatment does not reduce the proportion of people infective to mosquitoes, it decreases the proportion of mosquitoes that become infected after feeding on a treated individual [113], indicating the drugs have impacts upon post-transmission development. A remaining question therefore is why ARTs should be active against stages where no measurable Hb catabolism is occurring. Activity during exflagellation might be attributable to residual levels of heme, the resumption of low levels of Hb catabolism, or alternative MOAs all potentially amplified in gametogenesis by the sensitive signalling of this stage [71]. This current view also raises questions about the metabolism of *P. falciparum* and the MOAs of artemisinin derivatives.

## 4. Potential Directions Based on Gametocyte Biology

### 4.1. Cytoskeleton Remodelling

Following the infection of erythrocytes by committed merozoites, early gametocytes are morphologically indistinguishable from the ABS. During progression from stages II to IV, the infected cell increases in size and becomes more rigid. Whereas changes in ABS morphology are attributable to remodelling of the host erythrocyte cytoskeleton, the morphological maturation of the gametocyte is mediated by changes to the parasites' own cytoskeleton and inner membrane complex (IMC) [114]. It is hypothesised that a major reason for such dynamic regulation of cytoskeleton rigidity is to promote the retention of the developing immature gametocyte within the bone marrow [115,116], and remodelling to a deformable phenotype at stage V is suggested to be enable their escape to the circulation [117]. Interventions that take advantage of these profound changes in deformability may represent a novel approach to target the gametocyte and interrupt transmission [118,119].

Gametocytes are reliant upon STEVOR (subtelomeric variable open reading frame)-mediated interactions between the erythrocyte plasma membrane and the erythrocyte ankyrin complex to modulate their morphology, the regulation of which is controlled by protein kinase A (PKA) cyclic AMP (cAMP)-dependent phosphorylation [120]. PfPKA mediated phosphorylation of STEVOR promotes interactions with ankyrin complex to increase cellular rigidity in stage III and IV. cAMP degradation by parasitic phosphodiesterases (PfPDEs) regulates cAMP levels, and a threefold increase in the expression of PfPDEδ by the stage V compared to III, and corresponding fivefold reduction in intracellular cAMP, appears to produce the loss of rigidity at stage V [121].

In some of the first data that showed this could represent a novel gametocyte-specific target, the inhibition of PfPDE with sildenafil reverted the mature gametocyte to its earlier rigid morphology via increased intracellular cAMP levels, and the altered morphology impeded travel through an in vitro splenic clearance model [121]. The treatment of *P. berghei*-infected mice with sildenafil resulted in an accumulation of gametocytes in the bone marrow and spleen [122], though it will be important to confirm these findings in *P. falciparum* gametocytes, from which they are radically different. Advances in producing a humanised mouse model that supports a *P. falciparum* infection [123] and improved HTS protocols to assay gametocyte deformability [124] might enable these questions to be answered, as well as to identify additional compounds which interrupt these processes. Indeed, compounds not known for their ability to modulate gametocyte morphology are reported to cause retention in splenic clearance models [125].

Whether mature gametocytes preferentially accumulate within the capillaries of the dermis to facilitate increased mosquito uptake remains an unresolved question [126,127]. It has been hypothesised that their potential accumulation in the dermis is reliant on their crescent shape [128], such that they are oriented within the blood flow to facilitate retention in these small capillaries. This would suggest that interfering with their morphology could hinder dermis localisation and thus their uptake into the mosquito. Disrupting these processes may therefore not be lethal to the parasite, but inhibit escape from the bone marrow, splenic passage, and accumulation in the dermis.

### 4.2. TCA Cycle Dependence

Whereas the asexual parasite's energy needs are predominately met by cytosolic anaerobic glycolysis, the gametocyte is dependent on a switch to the tricarboxylic acid cycle (TCA cycle) to generate ATP [129]. Lacking several enzymes required for gluconeogenesis, the parasite must source all of its glucose from the host circulation [130]. The gametocyte's removal from the circulation and need to survive in the nutrient-poor mosquito may be the causes for this switch in metabolism [131,132]. Knock out of TCA cycle enzymes does not inhibit ABS growth but impairs the sexual stages either as gametocytes or during the mosquito sexual stages [132], and gametocyte expression of 15 of 16 mRNA transcripts encoding TCA cycle enzymes is increased compared to the ABS [133]. Isotope metabolite labelling and pharmacological inhibition studies demonstrate the high flux of the pathway and its essentiality for gametocyte development [49,52,129].

Currently, there are a lack of suitable compounds that block transmission by inhibiting enzymes of the TCA cycle. Compounds that specifically inhibit enzymes of the parasites' TCA cycle may be effective gametocytocidals, and characterisation shows at least one, fumarate hydratase, has significant differences to the human analogue, which might represent a possible target [134]. In addition to considering specific TCA enzymes, the parasitic mitochondrion may similarly represent an attractive drug target.

### 4.3. Mitochondrial Activity

*P. falciparum* merozoites inherit a single mitochondrion, which in the ABS remains small and, while necessary to numerous metabolic processes, is not required for the generation of ATP [135]. Within the gametocyte, this organelle undergoes marked expansion, developing into a large and elongated form with visible cristae [136,137]. This is accompanied by a significant upregulation of transcripts encoding mitochondrial proteins [133]. As discussed above, the gametocyte relies on the mitochondrial function to generate ATP via the TCA cycle. Additionally, the post-fertilisation stages within the mosquito, the zygote and ookinete, rely on oxidative phosphorylation to produce energy [138–140]. Mitochondrial inhibitors might, therefore, be expected to be effective gametocytocidals; however, studies to date have produced contradicting results.

Both gametocytes and ABS parasites depend on the mitochondrial electron transport chain (mtETC) for pyrimidine biosynthesis [141]. Specifically, cytochrome bc1, the third complex of the mtETC, recycles ubiquinol to ubiquinone through action as an electron acceptor. Given that ubiquinone is an essential coenzyme for pyrimidine biosynthesis, this represents a critical role for cytochrome bc1 [142].

Atovaquone is a well-characterised mitochondrial inhibitor currently used to treat malarial infections. Atovaquone is a ubiquinone analogue and a potent and selective inhibitor of the *P. falciparum* cytochrome bc1 complex. This inhibition not only causes a collapse in the mitochondrial membrane potential but indirectly inhibits pyrimidine biosynthesis by depriving the parasite enzyme dihydroorotate dehydrogenase (PfDHODH) of its cofactor ubiquinone [143]. Atovaquone kills the ABS relatively slowly by restricting nucleotide availability [144]. Despite earlier results reporting that atovaquone was active against the gametocyte [145], the consensus is that atovaquone is not directly gameto-cytocidal [64,70,87,111]. Atovaquone has no activity against female gametogenesis but may be active against male gametogenesis [71,73]. The transmission blocking activity of atovaquone is well-described, where it inhibits ookinete formation and reduces oocyst counts [146]. Its comparable activity in direct and indirect SMFA assays [74] and ability to block transmission in mosquitos that make contact with atovaquone treated surfaces [147] indicate that activity against the mosquito stages is critical.

The lack of consistently observable activity against exflagellation is surprising given their need for nucleotides for DNA replication. Microgametes, however, derive their energy from glycolysis [148]. Macro-gametogenesis does not require immediate DNA replication, though the survival of the fertilised cell requires a functioning mitochondria [54]. It therefore appears that atovaquone acts primarily against the post-fertilisation stages by inhibiting the mtETC of fertilised gametes. Atovaquone has a long half-life in serum of 2–3 days in adults [149], and sera from treated patients can block transmission for over 35 days after treatment [150], suggesting this as a strategy to inhibit transmission.

An interesting comparison to atovaquone is the compound DSM265, which directly inhibits PfDHODH and has shown potent activity against the ABS in vitro [151] and in clinical trials [152,153]. Like atovaquone, DSM265 does not kill immature or mature gametocytes, or inhibit male or female gametogenesis, but does decrease oocyst counts [154]. Similarly, it does not clear circulating gametocytes in clinical trials [155,156]. The activities of both atovaquone and DSM265 are therefore restricted to the ABS and post-fertilisation stages. These findings suggest that either pyrimidine biosynthesis and/or the mtETC are less important for gametocyte development than other lifecycle stages. The potential of the mtETC as a suitable gametocyte-specific target therefore remains an open question.

*4.4. Lipid Metabolism*

Lipid molecules have diverse roles in regulating membrane properties, signalling pathways, and energy metabolism. The characterisation of the lipidome profile of game-tocytes and the ABS stage reveals marked differences in lipid content and in total lipids, between ABS and gametocyte stages [157]. Stage V gametocytes contain twice the total amount of lipids compared to trophozoites, with marked differences in lipid content. For example, the proportion of phospholipids is nearly halved in stage V gametocytes relative to the trophozoite, whereas neutral lipids and cholesterol are increased significantly. Lipid composition additionally varies between male and female gametocytes, and the inhibition of lipid metabolism might therefore represent a strategy to target these stages [158]. Several of the enzymes utilised during lipid metabolism by *P. falciparum* are sufficiently distinct from or absent in humans [159]. Two compounds that interfere with ceramide synthesis, GT11 and MSDH-C, have shown gametocytocidal activity [160]. Lipid-adjacent targets could also be inhibited in gametocytes. PI(4)K is a kinase that phosphorylates lipids to regulate intracellular signalling and trafficking, and the PI(4)K inhibitor KDU691 was gametocytocidal, inhibited gametogenesis, and blocked transmission in the SMFA [161].

While growing in the erythrocyte, ABS parasites cause a breakdown in lipid asymmetry between the inner and outer membrane leaflets that is maintained in healthy uninfected erythrocytes [162]. Parasite infection causes loss of asymmetry, characterised by increased negatively charged lipid phosphatidylserine in the outer leaflet. This contributes to the increased adhesiveness of parasitised cells, important for immune evasion, and opposingly promotes phagocytosis by macrophages [163]. Molecules sensitive to the changes in external membrane character of infected erythrocytes are therefore a potential therapeutic avenue. Of promise are antimicrobial peptides (AMPs) such as those derived from platelet factor 4 (PF4), the effector molecule of human platelet anti-plasmodial activity [164,165]. Synthetic mimetics of the PF4 AMP domain are active against ABS parasites in vitro [166], and peptides of stabilised dimers of the PF4 AMP domain are selectively internalised by infected but not healthy uninfected erythrocytes. Accompanying biophysical studies of these peptide constructs using synthetic vesicles and membranes with defined lipid compositions reveals their specificity is dependent on negatively-charged phospholipids, similar to the phenotype of an infected erythrocyte [167]. The importance of membrane maintenance in gametocytes or indeed their sensitivity to such compounds remains to be investigated.

## 5. Targeting Gametogenesis and the Post-Fertilisation Stages

Stage V gametocytes are primed for explosive development in the mosquito midgut, and the energy requirements of gametogenesis necessitate an active metabolic state, akin to the ABS in several respects [168]. Male gametogenesis is particularly energy-intensive, reflecting the need for motility and rapid nuclear replication [169]. Female gametogenesis does not require genomic replication but instead involves the translation of previously repressed mRNA transcripts generated by the female gametocyte and stored as messenger ribonucleoproteins (mRNPs) [170]. Though the exact mechanisms critical to regulating zygote and ookinete maturation remain largely unknown [171], the survival of the parasite in the midgut and its escape to the haemocoel are dependent on the expression of these maternally derived mRNPs [172]. These processes are only usefully targetable if the drug treatment is able to pass from treated patient into the mosquito at levels sufficient to be effective, or if they are abrogated within the gametocyte prior to transmission.

### 5.1. Folate Dependent Pyrimidine Biosynthesis

Folic acid is a necessary cofactor for de novo pyrimidine biosynthesis and requires recycling for continued use. In *P. falciparum*, the enzyme dihydrofolate reductase (PfD-HFR) catalyses the regeneration of tetrahydrofolic acid from dihydrofolate, recycling this compound for use in pyrimidine biosynthesis [173,174]. Cycloguanil and pyrimethamine inhibit PfDHFR and have been deployed against malaria since the 1940s [175]. Their molecular MOA is well characterised, whereby they competitively inhibit PfDHFR through binding to its active site [176].

There is no activity of either compound against any gametocyte stage [64]. However, both drugs inhibit male but not female gametogenesis [71,73] and reduce oocyst counts [177]. The controlled timing of pyrimethamine ingestion shows that uptake by the mosquito is critical for their transmission blocking activity, as drug ingestion 2–4 days after an initial infective meal does not inhibit oocyst counts [177], and the washing of pyrimethamine from exposed gametocytes prior to feeding restores them [74].

The stages most susceptible to cycloguanil and pyrimethamine, the ABS and activating male gametocytes, both actively perform DNA synthesis and replication, whilst gametocytes and macrogametes do not. This is consistent with a mechanism that deprives the parasite of the pyrimidine nucleotides required to sustain replication. How can this activity profile be reconciled with that of atovaquone and DSM265, which also interfere with pyrimidine biosynthesis, but do not inhibit exflagellation? Perhaps additional MOAs, the redundancy of the enzymatic functions, or temporal inhibitory effects are the basis for variation between these drug classes. An increased understanding of the underlying biological processes would allow for reconciling this difference in activity.

*5.2. Gametogenesis Signalling Pathways*

Regulation of commitment to gametogenesis relies on sensitive signalling pathways that ensure rapid activation within the mosquito midgut but not within the vertebrate circulation. Two key signalling molecules in gametogenesis are cGMP and calcium, which stimulate downstream cascades that initiate and regulate gametogenesis [178]. These molecules are also critical to the merozoite and sporozoite, the other cell-invasive stages of *Plasmodium* [179–181]. In *P. falciparum*, cGMP levels are regulated by two cGMP-producing guanylyl cyclases and four cGMP-degrading PfPDEs [182,183]. Following mosquito ingestion, and at permissive temperatures, xanthurenic acts as a natural agonist of guanyl cyclase to increase cGMP levels within the parasite [32]. The primary, and perhaps only, effector protein of cGMP in *P. falciparum* is a cGMP-dependent protein kinase (PfPKG), which has essential roles across several lifecycle stages [183]. PfPKG controls the release of parasitic calcium stores, which activate stage-specific calcium dependent protein kinases (PfCDPKs), which, in turn, facilitate much of the activity of gametogenesis [184,185].

A number of PfCDPKs together regulate the processes of exflagellation [184]. PfCDPK4 is essential to initiate genomic replication, cellular rearrangement, and axoneme assembly [185]. A second kinase, PfCDPK1, appears to have roles in triggering gamete egress from the erythrocyte [186]. PfCDPK1 is the target of the compound purfalcamine, which, against the ABS, causes arrest at the late schizont stage [187], and inhibits microneme discharge and erythrocyte invasion by merozoites [188], perhaps suggesting a role in inhibiting exflagellation by this class of compound.

The inhibition of PfPKG in the ABS blocks merozoite egress from schizonts by inhibiting calcium release, while the PDE inhibitor zaprinast raises cellular cGMP levels and triggers premature escape from the erythrocyte [189,190]. PfPKG inhibition similarly blocks exflagellation and rounding out of activated gametocytes [34]. During the ookinete stage, motility and escape from the midgut are dependent on PfPKG mediated calcium release [191]. Several PfPKG inhibitors are under investigation [192]. For example, MMV030084 inhibition of PfPKG blocks sporozoite invasion, ABS growth, and male gametogenesis [193]. The shared nature of PfPKG-dependent activation of PfCDPKs highlights the potential for targeting this process.

*5.3. Female Translational Repression*

Translational repression is a mechanism of post-transcriptional gene regulation used extensively by *Plasmodium*. Indeed, the *P. falciparum* genome contains a relatively high proportion of RNA binding proteins compared to other eukaryotes [194,195], with potential roles across the lifecycle [196,197] and appears especially critical for female gametogenesis and post-fertilisation development [198]. These mechanisms demonstrate the "long arm" of the gametocyte's early preparations for transmission and the potential for far reaching effects on transmission biology by drugs administered to the gametocyte.

Several gene transcripts associated with ookinete development are expressed by immature female gametocytes and stored as mRNPs [199]. The release in translational repression occurs in response to entering the mosquito midgut environment, by fertilisation by the microgamete, and other factors that remain unknown [200,201]. The analysis of the constituent proteins of mRNPs has identified several important for sexual development including Pf-DOZI (development of zygote inhibited), PfCITH (homolog of CAR-I and fly Trailer Hitch), PfPUF2 (Pumilio/FBF 2), and others in both *P. falciparum* and *P. berghei* [202–204]. PfDOZI or PfCITH knockout permits normal ABS and gametocyte development; however, fertilised cells are unable to develop to mature ookinetes [205]. Approximately 700 gene transcripts are associated with PfDOZI and PfCITH [206,207].

There do not appear to be drugs that directly interfere with these mRNP complexes or their function. However, compounds that interfere with these complexes or disturb the balances that control translational repression may offer opportunities to render the gametocyte sterile or nonfunctional in the mosquito. These could include compounds that inhibit nucleotide biosynthesis, or those that interact with the proteins of mRNPs. Such activity could have downstream effects on ookinete development, as evidenced from gene knockout experiments. The mechanisms of translational repression, as deployed uniquely in *Plasmodium*, remain to be fully characterised, but represent the diversity of gametocyte biology and the variety of molecular processes that might be targeted in the future.

## 6. Future Perspectives

Undoubtedly, much remains to be known regarding the biology of the gametocyte, its activation, and its progression through the post-fertilisation stages in the mosquito. Insights into these areas will enable the identification of new druggable targets. Drugs developed for use against the ABS often have incidental activity against the gametocyte; indeed, "antimalarials" are so named because of their activity against the symptomatic ABS, whereas gametocytes contribute minimally to pathogenesis [15]. These different activity profiles may be partly explained by the differences in metabolism required for survival in the bone marrow and mosquito, especially in haemoglobin catabolism, mitochondrial activity, and nucleotide biosynthesis. Processes that are unique to the gametocyte from the ABS, such as the dependence on the TCA cycle, increased mitochondrial development, cytoskeletal regulation, and lipid metabolism might represent avenues for specific gametocytocidal therapies.

Targeting the post-fertilisation stages is a further possibility, either via drug uptake in the blood meal, or by attenuating critical processes in the gametocyte prior to transmission. This, however, is complicated by the need to ensure adequate drug uptake in the blood meal, or continued effects against the gametocyte, and might best suit compounds with long half-lives [208]. Gametocytes circulate in the blood for several weeks, though this can be reduced fourfold to a carriage time of two weeks following ART treatment [29]. A gametocytocidal compound would therefore need to persist in circulation for at least this period. A more complete understanding of post-transmission biology, including in those processes described in this review and other targets that remain as yet unknown, will aid in realising such a strategy [209].

What further lessons can be applied to efforts to develop gametocytocidals? The need for cellular and molecular biology to direct treatment strategies is clear, and the intelligent selection of partner drugs with complementary MOAs will be critical for next generation antimalarials [210]. Specific gametocytocidals in combination with partner ABS treatments could be especially useful in limiting drug resistance, as current ABS drugs with partial gametocytocidal activity have resulted in reduced sensitivity in the sexual stages they additionally target [211,212]. Estimates based upon the number of ABS parasites in circulation and the average rate of nuclear mutation predict as many as 1 in 50 ABS parasites contain a novel mutation [78]. From a total ABS population of $10^8$–$10^{12}$ parasites, this represents a significant opportunity for resistance to occur. In contrast, gametocyte

numbers are reduced by several orders of magnitude, and even fewer are transmitted, perhaps reducing opportunities for resistance to specific gametocytocidals to emerge [213].

In looking to the future of gametocytocidal therapies, a consideration of the manner and scope of their use should be given. Therapies given solely to disrupt transmission will provide no direct benefits to the patient save for the decreased risk of possible reinfection. They may, therefore, be unwarranted in high transmission areas [214], or where a strong rationale to prevent transmission from symptomatic individuals is lacking. Currently, primaquine is the only gametocytocidal recommended by the WHO, often cited to highlight the dearth of treatments that interrupt transmission [215]. In practice, however, the WHO's recommendation is far more limited than universal use, recommending primaquine to limit the spread of ART resistance in areas of low transmission [216]. The prescription of gametocytocidal therapies to shield against resistance emergence or spread could therefore represent a practical use case [217,218].

If the goal of gametocytocidal therapies is instead to control malaria at the population level, then all carriers of gametocytes will have to be identified and treated. In malaria endemic regions the largest carrier population of gametocytes are not symptomatic patients but asymptomatic individuals [219], and transmissible gametocyte densities can be below the threshold of field detection methods [220]. A mass drug administration (MDA) campaign to clear all symptomatic, asymptomatic, and (without sensitive screening technologies) uninfected individuals of their invading gametocytes and ABS parasites would be necessary [221,222], requiring high participation rates and presenting a high barrier to success [223,224]. Understanding the rationale for specific transmission blocking therapies additionally requires knowledge of the complex lifecycle of the malaria parasite, and may unnecessarily increase the reliance of individuals without that knowledge on their health care providers [225]. *P. falciparum* has already been eradicated from large areas of the world, including several countries in the twenty-first century, with variable dependence on primaquine use [226], and a failed MDA could theoretically cause harm if it contributed to drug resistance [227]. The urgency of the effort to control malaria suggests a policy of proactivity in this area [228]. Nevertheless, we are faced with the question of whether individuals should be given or be required to take medicines that will not directly benefit them. Ultimately, these questions will need to be answered by the communities, clinicians, and individuals of world's malaria endemic regions.

It will hopefully remain a strong priority to continue to understand the gametocyte and its transmission, with the prospect that this information will lead to the identification of new druggable targets. The challenges now lie in continuing to unravel the biology of this key bottleneck in the *Plasmodium* lifecycle and in translating this growing inventory of information into practical data and safe and effective treatments; therapies that will likely change malaria treatment as it is currently understood. In this review, we have laid out some strategies to direct this study and highlighted progress in this fascinating area of malaria research.

## 7. Concluding Remarks

Despite decades of research, much of the biology of *Plasmodium* remains to be elucidated. Uncovering how this species can transform itself from the proliferative asexual parasite, through the extended process of gametocytogenesis, and into the uniquely adapted sexual stages of the mosquito midgut will provide information on the mechanisms of antimalarial activity and help to develop new strategies to elevate the world's malaria burden. The gametocyte represents a stage that could be uniquely vulnerable to drug action, both in its biology and lifecycle juncture, and should continue to be an area of focus in developing future antimalarials.

**Author Contributions:** Conceptualization, B.A.M. and B.J.M. Writing—original draft preparation, B.A.M. Writing—review and editing, B.J.M. Supervision, project administration, and funding acquisition, B.J.M. All authors have read and agreed to the published version of the manuscript.

**Funding:** This research was funded by National Health and Medical Research Council, Australia, grant number GNT1183927.

**Institutional Review Board Statement:** Not applicable.

**Informed Consent Statement:** Not applicable.

**Data Availability Statement:** Not applicable.

**Acknowledgments:** We thank the Australian National University for scholarship support for B.A.M., and funding from the National Health and Medical Research Council (GNT1183927).

**Conflicts of Interest:** The funders had no role in the design of the study; in the collection, analyses, or interpretation of data; in the writing of the manuscript, or in the decision to publish the results.

## Abbreviations

| | |
|---|---|
| ABS | Asexual blood stage |
| AMP | Antimicrobial peptide |
| ARTs | Artemisinin derivatives |
| ATP | Adenosine triphosphate |
| cAMP | Cyclic adenosine monophosphate |
| CDPK | Calcium-dependent protein kinase |
| cGMP | Cyclic guanosine monophosphate |
| CITH | CAR-I/Trailer Hitch |
| DHA | Dihydroartemisinin |
| DHFR | Dihydrofolate reductase |
| DHODH | Dihydroorotate dehydrogenase |
| DOZI | Development of zygote inhibited |
| DV | Digestive vacuole |
| G6PD | Glucose 6 phosphate dehydrogenase |
| Hb | Haemoglobin |
| HTS | High throughput screen |
| IMC | Inner membrane complex |
| MB | Methylene blue |
| MDA | Mass drug administration |
| MMV | Medicines for Malaria Venture |
| MOA | Mechanism of action |
| mRNA | Messenger ribonucleic acid |
| mRNP | Messenger ribonucleoprotein |
| mtETC | Mitochondrial electron transport chain |
| NADPH | Nicotinamide adenine dinucleotide phosphate |
| PDE | Phosphodiesterase |
| PF4 | Platelet factor 4 |
| PI(4)K | phosphatidylinositol-4-OH kinase |
| PKA | Protein kinase A |
| PKG | Protein kinase G |
| PQ | Primaquine |
| PUF2 | Pumilio/FBF 2 |
| SMFA | Standard membrane feeding assay |
| STEVOR | Subtelomeric variable open reading frame |
| TCA cycle | Tricarboxylic acid cycle |
| WHO | World Health Organisation |

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
