# Peer review of "Antimalarial Drug Strategies to Target Plasmodium Gametocytes"

_parasitologia, doi:10.3390/parasitologia2020011_

Round 1
Reviewer 1 Report
The present review aims at summarizing present and future transmission blocking treatments to control and eventually eliminate malaria.
While the second part related to potential new drug targets within the transmissible malaria stages both in human and in the mosquitoes, seems relatively well organized and complete, the first part of the review related to in use drugs and their mechanism of action needs clarification and a more careful quoting of the literature. More figures or schemes would help the reader.
In yellow in the manuscript, I outlined the sections where modifications are suggested, whereas the detailed comments can be found below.
Line 45 check spelling
Line 142 I would call this paragraph Gametocytocidal Drugs or Compounds, but not therapies since only Primaquine at present is officially indicated by WHO as transmission blocking drug.
Line 151 please consider to add additional references on the screening of the MMV Malaria Box compounds, in particular the paper by Van Voorhis WC, Adams JH, et al. (2016) Open Source Drug Discovery with the Malaria Box Compound Collection for Neglected Diseases and Beyond. PLoS Pathog 12(7): e1005763. doi:10.1371/journal.ppat.1005763 , which summarizes all the data and the assays utilized by several laboratories to investigate the gametocidal activity of Malaria Box compounds.
Lines 156-158 same as above
Lines 193-97 Seminal work on hemozoin structure and chloroquine heme-binding by TJ Egan has not been quoted. Hemozoin is not a polymer and the existence of heme polymerase has never been confirmed at the molecular level. This must be corrected.
Lines 258-260 again references about different hypothesis on the mechanism of action of MB should be quoted ( see Haynes RK et al , 2010-2013 )
Line 814 please check characters
Line 926, reference 135 Vos et al , this reference should be added to the paragraph discussing SMFA ( lines 162-165) . The group of Koen J. Dechering made tremendous contribution to the SMFA assay
Line 928, reference 136 Lucantoni et al this reference should be added to the paragraph discussing the phenotypic screening methods for transmission blocking drugs (lines 156-158)

Reviewer 2 Report
The paper addresses a very current topic of great interest in the study of malaria control. It is very well structured in its sections, easy to read and extremely didactic. However, the bibliographic support must be updated, especially as it is a review paper.
I have nonetheless a few small suggestions:
line 13 “… areas of gametocyte are viable” with “areas” the authors mean “metabolic pathways” “biological processes”? Please clarify.
line 15 “parasite” or “gametocyte” biology?
lines 114 -116 - the content of these text is not entirely accurate, experimental infections reach much higher numbers of oocysts, see example in doi: 10.3389/fcimb.2021.634273 and many other works.
line 121 - this is a review paper. Please consider including at least one up to date reference to support the “genomic, transcriptomic, proteomic, and metabolomic-type studies”.
lines 119 – 141 Only 5 different references are provided to support the section “1.2. gametocyte biology” and they are largely out dated.
lines 167 - 169 please provide references to support this text.
line 209 - “… similarly restricted activity immature gametocyte …” there is a word missing?
line 209 -210 - provided references are out dated, see for example DOI: 10.1186/s12936-021-03706-1.
Line 355 – 356 – it is inaccurate that mitochondria is “… inactive during the ABS…” please see doi: 10.15252/msb.202010023.
References are particularly out dated in section 4.3 Mitochondrial Activity. For instance, the ones regarding atovaquone are from 1995 to 2013, so largely out dated. Please update them.
Line 426 – Lacks bibliographic support for DSM265, see DOI: 10.1093/jac/dkab181 or DOI: 10.3390/ijms22137236.
Line 519 - “(10% of annotated proteome)” is this relative to P. falciparum or other eukaryotes? Please rephrase in order to clarify.
Line 532 - references are from 2006 – 2014, so out dated see DOI: 10.1371/journal.ppat.1007249.
Line 538 – please update references see DOI: 10.15252/embr.202051660, DOI: 10.1186/s13072-021-00393-9 or DOI: 10.1111/mmi.14334.
Line 541-548 – Speculative and broad, please name the compounds.
Line 549 – 629 - the section “6. Future Perspectives” provides excellent ideas, however is to extensive and speculative it needs some well-balanced bibliographic support.
Reviewer 3 Report
The aim of the manuscript “Antimalarial drug strategies to target Plasmodium gametocytes” is to review the data on activity of the current repertoire of antimalarials on P. falciparum gametocytes and to propose how gametocyte biology can provide opportunities to identify targets and to develop novel molecules active against the human to mosquito parasite transmission stages.
Unfortunately the manuscript does not reach these objectives and despite the use of a vast, but sometimes outdated and incomplete, bibliography
1- limits analysis to three licenced drugs and does not discuss the most recent development in the identification of compounds active against gametocytes and their proposed mechanisms of action,
2- dedicates most of the attention to processes and molecules active on the parasite stages developing in the mosquito rather than those relevant to the blood stage gametocytes, minimally discussing the drug development implications of targeting gametes and/or sporogonic development,
3- conveys several inaccurate pieces of information on parasite and gametocyte biology, which is inacceptable for a high profile parasitology review
4- ends with a Future Perspective section that recapitulates a variety of issues on the feasibility and the obstacles in developing malaria parasite transmission blocking strategies, without a clear focus and message, beside that of the importance of elucidating mechanisms of gametocyte biology.
Observations motivating this opinion and specific points on the manuscript style are the following.
line 11: the adjective “innate” is used next to “gametocyte biology” throughout the manuscript to indicate what in fact is “gametocyte biology”. Is there a “non innate” gametocyte biology?
line 13: drug targets can be molecules or processes, not “areas” of the gametocyte.
line 16: the term “story” is unclear in this sentence.
lines 32-34: the meaning of the sentence “The adaptations..” is obscure.
line 37: “rapidly” is meaningless, unless it is compared to some “slowly” dividing stages this adverb seems to refer to.
line 38: asexual blood stage is not “where” malaria symptoms occur.
lines 44 and 46: correct “gametocyte” to match with “their”.
lines 48-51: the sentence is poorly written; the niches are not indicated and the meaning is obscure.
line 53: sporozoites do not enter circulation during the blood meal; the review should mention the work describing the interactions of these stages in the derma prior to accessing the microvasculature.
line 60: “invasion and escape” (egress would be a better term than escape) should be followed “of and from erythrocytes, respectively”.
line 63: stating that schizont rupture is responsible for most clinical symptoms of malaria fails to describe major processes in malaria pathogenesis (e.g. parasite cytoadherence and the consequences on the affected organs).
line 69: the indication of the factors driving commitment to gametocytogenesis is inaccurate and outdated, the references chosen are marginal. The indication that “high parasitaemia” is one of these factors derives from laboratory observations on cultured parasites, very poorly translatable to the situation of natural infections.
lines 71-73: the work and conclusions of reference n. 20 have been completely misinterpreted.
line 85: reference n. 28 is a clinical case observation; that report was followed by more extensive observations which deserve being quoted. The “other hematopoietic niches”, where gametocytes are here described to home and mature, should be mentioned, with appropriate references.
line 130: the indication that “intracellular cellular structures” are shared by asexual stages and immature (please notice that the term immature indicates stages from I to IV) requires some detail.
line 147: gametocyte refractoriness to many antimalarial drugs active on asexual stages is the subject of several recent studies which deserve being quoted here, in addition or in alternative to reference n. 50.
line 148: the statement that many treatments for malaria stimulate gametocytogenesis is an oversimplification and is partly inaccurate. Clinical observations that a wave of gametocytaemia can follow drug treatment, particularly in those described in reference n. 53, cannot be explained by an increase in gametocytogenesis as the timing of appearance of gametocytes in circulation is incompatible with the P. falciparum long gametocyte maturation period.
lines 151-153: the sentence is confused and poorly written: gametocytes (these cells reside virtually only in the human host, except for the few minutes before activation to transform into gametes in the mosquito gut) are targeted because they are required for parasite transmission, not because of their location.
line 157: “dormancy” is normally applied to biological states very different from that of mature gametocytes. Reference quoted here, the comparative analysis in ref n. 62, does not define mature stages as dormant, but correctly uses the term metabolically “inactive”, referred to haemoglobin digestion.
line 165: The manuscript ignores recent developments that introduced the use of parasite bioluminescence as an SMFA readout.
line 186: the title Dead ends (plural) is not appropriate as only Haemoglobin catabolism is described in this section as a non practicable target to kill mature gametocytes.
Figure 1: the illustrations of asexual ring and trophozoites are placed in the same area with the “mosquito” colour code that indeed contains the illustrations of the parasite mosquito stages; this may be confusing for a non expert reader. In the same figure, the term “translational repression” placed in the box “immature gametocytes” is confusing and misleading.
line 221: The title “Known gametocytocidals” is inaccurate as only licensed drugs are described here. With this title, it would be expected that this section reviewed also results on several hits and leads which have recently emerged as efficient gametocytocidals from several HTS.
line 234: what are “both” parasite stages mentioned here?
line 239: The literature on methylene blue quoted in this paragraph is limited, ignoring articles more directly addressing MB activity in gametocytes (for instance, ref n. 81 should be quoted here).
line 297-300: Several errors are contained here: 1- the causative association between immature gametocyte rigidity and bone marrow localisation is highly speculative; 2- the increased deformability of mature gametocytes does not cause the crescent shape, this is shared with immature stages III and IV; 3- a role of the increased deformability of mature gametocytes in easing the release of these stages from the sequestration sites is more intuitive, although also this hypothesis is not directly demonstrated.
line 314: the adjective mature is missing before “gametocyte”.
line 316: Morphology, development/maturation and cell structure of P. berghei gametocytes are so different from those of P. falciparum that description of these results, if maintained, requires some qualification and discussion.
line 328: “HTS for gametocyte flexibility” uses a wrong term (flexibility) and is obscure and inaccurate.
line 354: in the section dedicated to gametocyte biology, two out of five items (4.3 and 4.4) are in fact proposing to develop drugs active against the mosquito stages. This is the case for the parts describing the effect of atovaquone and DMS265 on pyrimidine synthesis and/or the mtETC, and that on antifolate inhibition of DNA replication.
line 383: “reports”, not “report” as only one reference (n. 137, quite old, as others used in the subsequent paragraph) is mentioned that describes activity, at odd with recent reports on this subject.
line 385: complete genomic replication occurs in activated microgametocytes, not in the haploid microgametes.
line 465-472: the arguments in this part are confusing. “Several therapies that are potent against the ABS but not the gametocytes” include drugs and compounds with diverse MoAs and targets, so it is unclear how these are collectively explained by the single “switch to a high metabolic activity in the mature gametocyte”. Drugs used in the above mentioned therapies affect different processes such as male gamete exflagellation and DNA replication in sporogonic development.
line 471: gamete egress indeed exposes the surface of these parasites to the inhibitory activity of antibodies; using the same argument for (likely membrane permeable) drugs requires some further explanation.
line 560: “nucleotide biosynthesis”: it is not clear what is the point in common proposed here between the development of the non-dividing parasite in the bone marrow and that of the dividing parasite in the mosquito.
lines 573-576: sentence is obscure.
Round 2
Reviewer 1 Report
The manuscript has been improved both in the description and in the accuracy of the references. In my opinion, it is now acceptable for publication
Author Response
Reviewer 1 has made no requests to revise the manuscript.
Reviewer 3 Report
The revision has improved the manuscript.
As images are more powerful than words to convey messages, Figure 1 requires further correction: the ookinete must be confined in the mosquito area (in the current version it looks like a human host invasive stage) and a break is needed between the mosquito area and the blood stage area to indicate that part of the cycle is missing here.
